# Reliability Evaluation of Hinged Slab Bridge Considering Hinge Joints Damage and Member Failure Credibility

**Guojin Tan** [1] , **Qingwen Kong** [1], **Longlin Wang** [2,]*, **Xirui Wang** [1] and **Hanbing Liu** [1]

[1]  College of Transportation, Jilin University, Changchun 130025, China; tgj@jlu.edu.cn (G.T.);
    kongqw18@mails.jlu.edu.cn (Q.K.); wangxr17@mails.jlu.edu.cn (X.W.); lhb@jlu.edu.cn (H.L.)
[2]  Guangxi Transportation Science and Technology Group Co., Ltd., Nanning 530007, China
*   Correspondence: wll955@163.com; Tel.: +86-1857-788-8161

**Abstract:** The hinged slab bridge is widely used in medium- and small-span bridges because of its simple structure and convenient construction. However, hinge joints damage is the main defect of this kind of bridge, and it is difficult to express the deterministic damage degree of hinge joints in the detection process. A system reliability evaluation method considering fuzzy detection information of hinge joints damage and member failure credibility is proposed in this paper. Firstly, the membership function is used to quantitatively express the fuzzy detection information of hinge joints, and the fuzzy variable is transformed to an equivalent random variable. Secondly, the functional relationship between the transverse distribution coefficient and hinge joints damage is constructed by the modified hinge-jointed plate method and response surface method, and the reliability of the member considering the fuzzy detection information of hinge joints damage is calculated by the first-order second-moment method (FOSM). Then, the failure credibility is introduced to represent the different possibilities of system failure caused by member failure, and a system reliability assessment method of different failure criteria considering member failure credibility is established based on copula theory. Finally, the applicability of the proposed method is verified by taking the reinforced concrete hinged slab bridge as a numerical example.

**Keywords:** hinged slab bridge; membership function; system reliability; failure credibility; response surface method; copula

## 1. Introduction

As the key node of road networks, bridge safety operation is very important to the normal operation of the transportation system. With the extension of the service period, the service function of the bridge will gradually deteriorate under the repeated action of the external environment and external load (such as floods, earthquakes, and vehicles, etc.) [1–7]. Therefore, in addition to optimizing the design of the bridge, it is of vital importance to the accurate judgment of the state of the bridge, which can not only provide a reasonable basis for the bridge maintenance decision, but also ensure its safe operation [8–14]. Hinged slab bridges are widely used in medium- and small-span bridges due to their simple structure and convenient construction, with one of the biggest defects being hinge joint damage, so it is of great significance to evaluate the status of hinged slab bridges under hinge joint damage [15–18]. As the material characteristics and external load have randomness, most commonly used probability-based assessment methods mainly include the first-order second-moment method (FOSM), Monte Carlo method (MC), and response surface method (RSM) [19]. The first-order second-moment method is based on the first-order function of the random variable in the structural function, and combines



the first-order moment and second-order moment probabilistic statistical parameters of the random variable to calculate the reliable index. For reliability analysis of a few dimensions and functions with clear expressions, the FOSM method is simple and efficient, which can usually obtain accurate results, [20]. Cho et al. used the FOSM method to calculate the reliability index of cracking at the upper and lower edge of the section of prestressed concrete box girder [21]. An et al. used the FOSM method to calculate the cable reliability of a suspension bridge under the effect of temperature [22]. The basic idea of the MC method is to carry out large-scale random sampling on random variables affecting the reliability of the structure, and substitute the sampling values into the structural function one by one to determine the failure times and calculate the failure probability. Zhu used the MC method to conduct a reliability assessment of the Zhaobaoshan cable-stayed bridge under the limit state considering the live load effect [23]. Gholipour evaluated the reliability of bending and shear failure of piers by MC simulation [24]. The MC method can avoid complex mathematical formulas in structural reliability analysis, but it has the disadvantage of an intensive calculation workload [25,26]. The clear functional expression cannot be obtained in the analysis of complex reliability, but the RSM can explicitly express the implicit functional function through a series of deterministic tests to fit an approximate limit state [27]. Ghosh adopted the adaptive RSM based on the least-squares method to carry out seismic reliability analysis of multi-span piers [28]. Yu conducted a reliability analysis by improved RSM on the limit state of typical self-anchored suspension bridges [29].

The above reliability calculation methods are all aimed at a single member or a single failure mode; however, the bridge structure is a complex structural system composed of multiple members, and the state of a single member cannot reflect the overall state of the bridge. Therefore, it is very important to evaluate the reliability of the bridge system. For a statically indeterminate bridge structural system, there are often multiple failure modes, but only a few major failure modes have great effect on the failure probability of the system. In addition, there is a certain correlation between members, and the load acting on the structure may also have the same random source, resulting in a correlation between the various failure modes. The recognition after the main failure mode of the structure system needs to consider the correlation of each failure mode, effectively integrating the failure mode to calculate the failure probability of the structural system reliability. After identifying the main failure modes of the structural system, it is necessary to consider the correlation of each failure mode to effectively integrate the failure probability of each failure mode to calculate the reliability of the structural system. Therefore, how to find the main failure modes and how to calculate the failure probability are two key problems in reliability analysis. In terms of structural failure mode search, the most commonly used methods are the optimality criterion method, β-Unzipping method, and branched-and-bound method [30]. Feng et al. proposed an optimality criterion method based on the load accumulation to search for major failure modes [31]. Lu et al. established a failure tree by the β-Unzipping method to identify the main failure modes [32]. Gao et al. proposed an improved critical intensity branched-and-bound method considering correlation between failure paths to identify major failure modes, which improved the search efficiency of traditional methods [33]. After the main failure modes are identified, the failure modes are usually combined by series and parallel systems to obtain the failure probability of the bridge structure by accurate calculation. In terms of calculating the failure probability of the system, the interval estimation method (mainly including the narrow bounds method and wide bounds method) and probabilistic network technique method are often used to calculate the failure probability of the system [34]. Cornell proposed the wide bounds method to calculate the failure probability of series and parallel systems, ignoring the correlation between failure modes, which could only roughly estimate the failure probability of structural systems [35]. Ditlevsen et al. proposed the narrow bounds method considering the correlation between any two failure modes, to determine the upper and lower bounds of the system failure probability [36]. Ang et al. arranged and grouped failure modes according to correlation based on the probabilistic network technique method, and selected m-group-significant failure modes to replace all the major failure modes to calculate the joint failure probability [37]. Although these methods have some applicability to some extent, there are some defects in the excessive

failure modes or the strong correlation between them. As Copula functions can not only capture the nonlinear correlation between failure modes, but also construct multiple joint distribution functions, which can easily calculate the joint failure probability of multiple members, they are introduced in the reliability assessment of structural systems. Jiang et al. established a theoretical model based on the Copula function and applied it in the reliability analysis of failure-related structural systems [38]. Liu et al. applied the Copula function to the system reliability assessment of simply supported beam bridges, and established the system reliability assessment method of series, parallel, and mixed models [39], which may be convenient and efficient for the structural system assessment of hinged slab bridges with multiple slabs.

In engineering practice, some parameters and detection information of bridge structures are difficult to be expressed by random variables, but these parameters and detection information have obvious uncertainty, so scholars introduce fuzzy theory into the risk assessment and reliability assessment of bridge structures. Jelena et al. proposed a bridge risk assessment framework under multiple disasters based on fuzzy theory [40]. Anoop et al. used fuzzy theory to evaluate the state of reinforced concrete bridges in a corrosive environment [41]. Wang et al. proposed a reliability assessment method for existing reinforced concrete bridges with incomplete test data based on fuzzy theory [42]. Adam et al. used the triangular membership function to characterize the fuzzy randomness of the interface stiffness of a composite beam bridge, and analyzed the dynamic response of the composite beam under a moving load [43]. Malekly et al. determined the most appropriate superstructure form in the bridge design stage based on the fuzzy comprehensive evaluation method [44]. Fabio et al. proposed a bridge reliability assessment method using interval-defined fuzzy criteria to deal with parameter uncertainties [45]. The application of fuzzy theory in bridge reliability assessment may not be sufficiently mature, and its application in hinge joint damage has not been found so far based on consulting relative literature.

A large number of systematic studies have been carried out on the structural reliability assessment of hinged slab bridges under the condition of hinge joint damage by the authors of this paper. A modified hinged plate method was proposed to calculate the transverse distribution of hinge joints after damage based on the traditional hinged plate method [46], and a reliability assessment method of the bridge system considering different failure criteria was proposed based on multiple Copula functions [47,48]. As the stress state of hinge joints is very complex, it is difficult for engineers and technicians to determine the damage degree of hinge joints without carrying out load tests, and they often give fuzzy information about the damage degree. The failure criterion of the existing system reliability assessment considers that the failure rate of the bridge structure due to each plate failure is equal, but the different slab failure contributions to the structure system failure are different because the slab space position is different. Therefore, how to effectively deal with the fuzzy information of the hinge joint damage degree and consider the different contributions of member failure to structural system failure are very important for the reliability evaluation of existing hinged slab bridge systems.

In view of the above problems, this paper originally proposes a novel method to evaluate the reliability of hinged slab beam bridges considering the fuzzy detection information of hinge joints damage and the failure credibility of members. The innovation of this paper is as follows: On the one hand, the membership function in fuzzy theory is used to represent the uncertainty of hinge joints damage degree detection information, and the membership function-equivalent randomization method to represent damage fuzzy detection information is proposed, which realizes the reliability calculation of a single slab considering hinge joints damage fuzzy detection information. On the other hand, the member failure credibility is used to characterize the different contributions of member failure to system failure, and the member failure credibility index is constructed. Then, a reliability assessment method for a hinged slab structure system is proposed combined with Copula theory. The specific process of the reliability evaluation method proposed in this paper is shown in Figure 1.

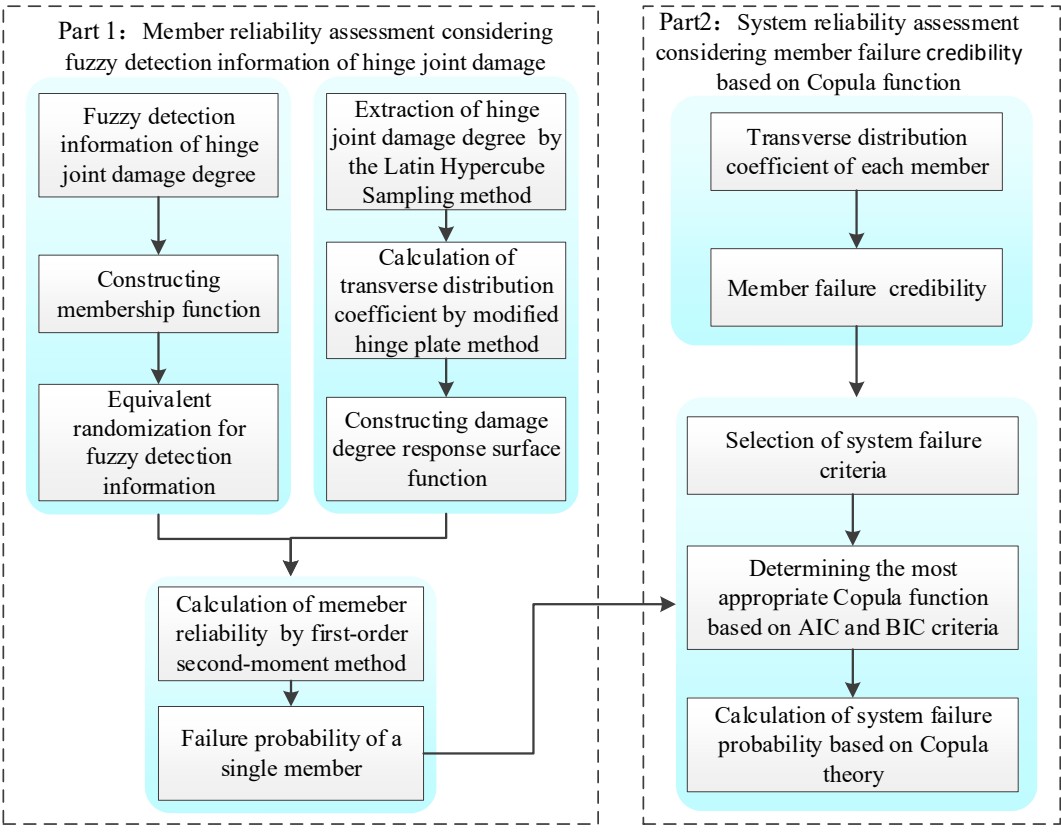

**Figure 1.** Flow chart of the reliability evaluation method proposed in this paper.

## 2. Member Reliability Evaluation Method of Considering Fuzzy Detection Information of Hinge Joints Damage

### 2.1. Fuzzy Detection Information Expression of Hinge Joints Damage Degree

The results of a large number of hinged slab beam bridges show that hinge joints damage is very common; however, it is difficult for engineers and technicians to quantify the damage degree in practical engineering deterministically due to the complex stress state of hinge joints. Maybe it is easier to give a fuzzy interval of the hinge joints damage degree. Therefore, the membership function of fuzzy theory is suggested to express the damage degree interval in this paper, in order to describe the recognition of the damage degree of hinge joints. At present, there are many types of functions that can be used as membership functions, among which the most commonly used types are normal, triangle, trapezoid, rectangle, and so on. The confidence degree of the middle values in the confidence interval of normal and triangle membership functions is the highest, while that of the upper and lower limits of damage is the lowest, and the confidence degree of the values in the damage interval is the same for rectangle type. The credibility of engineers in the damage interval is often inconsistent according to the law of credibility in practical engineering, and sometimes similar for each value in the interval. Therefore, triangle, normal, and rectangle-type membership functions are selected to express the damage interval [44]. $[D_L, D_U]$ is used to represent the damage degree interval given by the detection personnel, where $D_L$ represents the lower limit of the damage degree and $D_U$ denotes upper limit of damage. Therefore, triangle, normal-type, and rectangular membership functions are used to express the damage degree interval $[D_L, D_U]$ according to the concept of membership degree in fuzzy mathematics, which are as follows:

Triangle membership function:

$$u(\widetilde{x}) = \begin{cases} \frac{\widetilde{x}-D_L}{(D_L+D_U)/2-D_L}, & D_L < \widetilde{x} \le (D_L+D_U)/2 \\ \frac{D_U-\widetilde{x}}{D_U-(D_L+D_U)/2}, & (D_L+D_U)/2 < \widetilde{x} \le D_U \\ 0, & Others \end{cases} \quad , \tag{1}$$

where $\widetilde{x}$ is a fuzzy variable.

Normal-type membership function:

$$u(\widetilde{x}) = e^{-\left(\frac{\mu-\widetilde{x}}{\sigma}\right)^2}, \quad -\infty < \widetilde{x} < +\infty, \tag{2}$$

where $\mu = (D_L+D_U)/2, \sigma = (D_U-\mu)/k = (\mu-D_L)/k$, and $k$ represents the credibility of the detection personnel to the damage degree in the interval, and the greater the value, the greater the reliability.

Rectangular membership function:

$$u(\widetilde{x}) = \begin{cases} 1, & D_L \le \widetilde{x} \le D_U \\ 0, & Others \end{cases} \quad , \tag{3}$$

The above equation indicates that the credibility of detection personnel on damage degree in the interval is the same when the rectangular membership function is used.

However, the variables used in the reliability evaluation are all random variables, so it is difficult to directly solve the fuzzy reliability of the structure by using the membership function. Therefore, the method of equivalent randomization of the membership function is innovatively proposed in this paper to convert the fuzzy reliability calculation into an ordinary reliability calculation, and the fuzzy variables in the membership function are further transformed into random variables by obtaining the expression form of the probability density function, whose expression is as follows:

$$f(x) = \frac{u(\widetilde{x})}{\int_a^b u(\widetilde{x})d\widetilde{x}}, a < x \le b, \tag{4}$$

where $x$ is the random variable obtained from the transformation of fuzzy variable $\widetilde{x}$, and $f(x)$ is the probability density function of $x$, which satisfies the following conditions based on Equation (4),

$$\begin{cases} f(x) \ge 0 \\ \int_a^b f(x) = 1 \end{cases} \quad , \tag{5}$$

where $f(x)$ is in accordance with the nonnegative and normalization of probability theory; therefore, it is feasible to use this method to equivalently randomize the membership function.

The probability density functions of the three membership functions after equivalent randomization can be calculated according to Equation (4).

The probability density function obtained from the transformation of the triangle membership function is

$$f(x) = \begin{cases} \frac{2(x-D_L)}{(D_U-D_L)^2/2}, & D_L < x \le (D_L+D_U)/2 \\ \frac{2(D_U-x)}{(D_U-D_L)^2/2}, & (D_L+D_U)/2 < x \le D_U \\ 0, & Others \end{cases} \quad . \tag{6}$$

The probability density function converted from the normal membership function is,

$$f(x) = \frac{\frac{2k}{(D_U-D_L)/2}e^{-\left(\frac{(D_L+D_U)/2-x}{(D_U-D_L)/2}k\right)^2}}{1.775erf\left(\frac{(D_L+D_U)/2-\inf}{(D_U-D_L)/2}k\right) - 1.775erf\left(\frac{(D_L+D_U)/2+\inf}{(D_U-D_L)/2}k\right)}, -\infty < x < +\infty, \tag{7}$$

where $erf(\cdot)$ is the error function and inf expresses infinity.

The probability density function obtained from the transformation of the rectangular membership function is

$$f(x) = \begin{cases} 1/(D_U - D_L), & D_L \le x \le D_U \\ 0, & Others \end{cases}, \tag{8}$$

*2.2. Calculation of Transverse Distribution Coefficient Based on Damage Degree Response Surface Function*

The transverse distribution coefficient is an important parameter to measure how the load is distributed between the main beams. Therefore, the relationship between the degree of hinge joints damage and the transverse distribution coefficient is established to integrate the fuzzy information of hinge joints damage into the reliability evaluation. The Latin Hypercube Sampling (LHS) method is adopted in this paper to extract the damage degree samples of each hinge joint after the corresponding probability density function is obtained, and the modified hinge plate method and response surface method are used to calculate the transverse distribution coefficient, which is also a random variable. Taking the case of *N* main beams and *N*-1 hinge joints as an example, the specific steps of this method are illustrated as follows:

Step 1: Using the LHS method to extract samples of the damage degree of each hinge joint, each crack damage is considered as a random variable, each random variable is divided into n non-overlapping intervals within the range of [0,1], and the cumulative probability density function is obtained on the basis of the density function obtained by the equivalent randomization method of the random variable given in this paper.

Step 2: The modified hinge plate method proposed by the author of this paper [47] was used to calculate the transverse distribution coefficient of each main beam corresponding to the hinge joint damage degree of each set of sample points.

Step 3: The function relation between the transverse distribution coefficient and the damage degree of each hinge joint is constructed by using the response surface method according to the calculated sample point data, the selected response surface function is described in Equation (9), and the transverse distribution coefficient $m_i$ of each main beam considering the fuzzy detection information of the damage degree of hinge joints is obtained by using least-squares method.

$$m_i = a + \sum_{j=1}^{n-1} b_j x_j + \sum_{j=1}^{n-1} c_j x_j^2 + \sum_{1 \le j \le k \le n-1} d_{jk} x_j x_k \quad i = 1, 2, \cdots, n. \tag{9}$$

where $m_i$ represents the transverse distribution coefficient of the *i*-th main beam considering the fuzzy detection information of hinge joint damage degree; $x_i$ represents the random variable obtained from the transformation of the fuzzy detection information of the *i*-th hinge joint damage; and the remaining parameters are the undetermined coefficients.

*2.3. Reliability Assessment of Individual Members*

The general form of the limit state equation of each main slab is,

$$Z_i = R_i - SG_i - SQ_i \tag{10}$$

where $Z_i$ represents the limit state equation of the *i*-th member; $R_i$, $SG_i$, and $SQ_i$ represent the resistance, constant load effect, and live load effect of the *i*-th main beam, respectively.

The live load effect is calculated according to Equation (11):

$$SQ_i = (1 + \mu)m_i \times \overline{S}Q \tag{11}$$

where $\mu$ denotes the impact coefficient, $m_i$ is the transverse distribution coefficient of the *i*-th main slab, and $\overline{S}Q$ is the live load effect of the whole bridge under vehicle load, whose calculation equation is as follows according to China's General Specifications for Design of Highway Bridges and Culverts [49],

$$\overline{S}Q = q_k\Omega_k + p_k y_k \tag{12}$$

where $q_k$, $p_k$ refer to the uniform load and concentrated load of the vehicle, respectively. For simply supported slab bridges, $y_k = L/4$ and $\Omega_k = L^2/8$, where $L$ represents the calculated span of the bridge.

Combining Equations (10)–(12),

$$Z_i = R_i - SG_i - (1 + \mu)m_i(q_k\Omega_k + p_k y_k) \tag{13}$$

The FOSM method can be used to calculate the reliability of the single slab considering the fuzzy detection information of hinge joints damage.

## 3. System Reliability Evaluation Method Considering Member Failure Credibility

In the failure criterion of the existing system reliability assessment, it is considered that the contribution of each slab failure to the failure of the bridge structure system is equal. In fact, the remaining slab will bear the external load when a certain slab fails; however, the load effect of the remaining slab under the external load is also different when the failed slab is located at different spatial positions of the bridge. For example, the external load borne by the adjacent #2 and #3 slabs after the failure of the side slab is larger than that borne by the #2 and #3 slabs after the failure of the middle slab. Our research group concluded that different spatial positions of slabs in bridges would lead to different contributions of member failure to structural system failure of simply supported slab bridges through theoretical analysis and engineering practice. The engineers unanimously confirmed this inference after the investigation of experienced design engineers. Therefore, it can be considered that the contribution of different slab failures to the structural system failure of simply supported slab bridges is different; however, it is difficult to quantitatively express such differences by mechanical theory. This paper proposes to use reliability to express such differences. For hinged slab bridges, the transverse load distribution coefficient can reflect the different spatial positions of slabs and indirectly indicate the contribution of member failure to system failure. Therefore, the transverse distribution coefficient is adopted in this paper to construct the failure credibility index of hinge joints without damage, and its expression is as follows:

$$Bel_i = \frac{m_i^0}{\max(M)} \tag{14}$$

where $Bel_i$ is the failure credibility of the *i*-th member, $m_i^0$ represents the transverse distribution coefficient of the *i*-th member without damage, and $M = [m_1^0, m_2^0, \cdots, m_n^0]$ represents the vector composed of the transverse distribution coefficient of each main beam without damage.

The joint failure probability of the simultaneous failure of multiple members according to the Copula theory [48] is as follows:

$$P_f = P\{Z_1(X) \leq 0, Z_2(X) \leq 0, \cdots, Z_n(X) \leq 0\} = C\left(P_{fZ_1}, P_{fZ_2}, \cdots, P_{fZ_n}\right) \tag{15}$$

where $Z_i$ denotes the limit state equation of *i*-th members, $P_{f_{Z_i}}$ represents the failure probability of the *i*-th member, and $C(\cdot)$ expresses the copula function.

On this basis, member failure credibility is considered in this paper, and the joint failure probability of n members established by the Copula function is expanded in Equation (16).

$$
\begin{aligned}
P_{f,Bel} &= P\{(Z_1(X) \leq 0, Bel_1), (Z_2(X) \leq 0, Bel_2), \cdots, (Z_n(X) \leq 0, Bel_n)\} \\
&= C\left(P_{fZ_1,Bel_1}, P_{fZ_2,Bel_2}, \cdots, P_{fZ_n,Bel_n}\right),
\end{aligned}
\tag{16}
$$

where $P_{f,Bel}$ represents the joint failure probability of n members considering failure credibility, and $P_{f_{Z_i},Bel_i}$ represents the failure probability of the *i*-th member considering the failure credibility. As member failure probability and member failure credibility are independent variables, $P_{fZ_i,Beli} = P(Z_i(X) \leq 0) \times Bel_i$.

There are three kinds of hinged slab structure system failure modes: Failure criterion I considers that any one member failure causes the whole system failure; failure criterion II reckons that if two arbitrary adjacent members fail, the system fails; failure criterion III holds that if any three adjacent member fails, the whole system will fail. The authors of this paper proposed a structural system reliability assessment method considering three kinds of system failure criteria based on Copula function calculation [48]. Equation (16) is adopted in this paper to improve the system reliability assessment method previously proposed based on the failure credibility, and the system reliability evaluation method considering member failure credibility under failure criterion I, II, and III is formed, which are respectively derived as Equations (17)–(19).

In the case of failure criterion I,

$$
\begin{aligned}
P_{f_I,Bel} &= P\{(Z_1(X) \leq 0, Bel_1), (Z_2(X) \leq 0, Bel_2), \cdots, (Z_n(X) \leq 0, Bel_n)\} \\
&\approx \sum_{i=1}^{n} P\{(Z_1(X) \leq 0, Bel_1)\} - \sum_{1 \leq i < j \leq n}^{n} P\left\{(Z_i(X) \leq 0, Bel_i), \left(Z_j(X) \leq 0, Bel_j\right)\right\} \\
&\approx \sum_{i=1}^{n} P_{f_{Z_i},Bel_i} - \sum_{1 \leq i < j \leq n}^{n} C\left(P_{f_{Z_i},Bel_i}, P_{f_{Zj},Bel_j}\right)
\end{aligned}
\tag{17}
$$

where $P_{f_I,Bel}$ expresses the reliability of the system under failure criterion I considering member failure credibility.

In the case of failure criterion II,

$$
\begin{aligned}
P_{f_{II},Bel} &= P\{(Z_{11}(X) \leq 0, Bel_1), (Z_{12}(X) \leq 0, Bel_2)\} \cup P\{(Z_{22}(X) \leq 0, Bel_2), (Z_{23}(X) \leq 0, Bel_3)\} \cup \\
&\cdots \cup P\left\{(Z_{n-1,n-1}(X) \leq 0, Bel_{n-1}), (Z_{n-1,n}(X) \leq 0, Bel_n)\right\} \\
&= 1 - \prod_{i='}^{n-1}\left(1 - P\left\{(Z_{i,j-1}(X) \leq 0, Bel_{j-1}), (Z_{ij}(X) \leq 0, Bel_j)\right\}\right) \\
&= 1 - \prod_{i='}^{n-1}\left(1 - C\left(P_{f_{Z_{i,j-1}},Bel_{j-1}}, P_{f_{Z_{ij}},Bel_j}\right)\right), j = i + 1
\end{aligned}
\tag{18}
$$

where $P_{f_{II},Bel}$ expresses the reliability of the system under failure criterion II considering member failure credibility.

In the case of failure criterion III,

$$
\begin{aligned}
P_{f_{II},Bel} &= 1 - \prod_{i=1}^{n-2}\left(1 - P\left\{(Z_{ii}(X) \leq 0, Bel_i), (Z_{ij}(X) \leq 0, Bel_j), (Z_{ik}(X) \leq 0, Bel_k)\right\}\right) \\
&= 1 - \prod_{i=1}^{n-2}\left(1 - C\left(P_{f_{Z_{ii}},Bel_i}, P_{f_{Z_{ik}},Bel_k}\right)\right), j = i + 1, k = i + 2
\end{aligned}
\tag{19}
$$

where $P_{f_{III},Bel}$ expresses the reliability of the system under failure criterion III considering member failure credibility.

## 4. Numerical Simulation

In order to verify the correctness and applicability of the proposed reliability assessment method considering fuzzy detection information and member failure credibility, a typical expressway bridge was selected as the research object for numerical simulation analysis. The bridge is a single span-reinforced concrete hollow slab beam, which is composed of five hollow slabs. The total length of the bridge is 10.0 m and the total width of the bridge deck is 7.5 m. The main beam is constructed with concrete of grade C50, reinforced with HRB335 and R235 steel bars. The material properties of the bridge, as well as the statistical parameters and probability distribution types of random variables of main slab resistance, dead load effect, and live load effect, are the same as those of reference [47]. The profile diagram of the bridge and the location of hinges are shown in Figure 2.

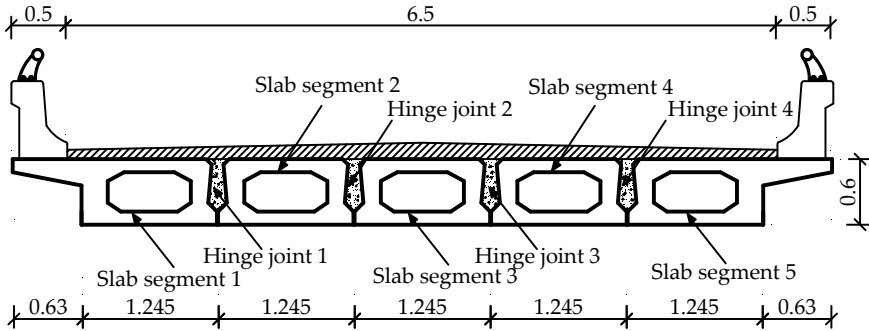

**Figure 2.** Bridge profile diagram.

### 4.1. Member Reliability Analysis

Four representative damage cases of hinge joints are selected on the grounds of the number of damaged hinge joints in this paper. Case 1: Hinge joint 1 damages; Case 2: Hinge joints 1 and 2 damage; Case 3: Hinge joints 1, 2, and 3 damage; Case 4: Hinge joints 1, 2, 3, and 4 damage. The difference between the cognition level and engineering experience of the detection personnel will lead to different fuzzy detection intervals; the smaller the fuzzy detection interval, the higher the confidence degree of hinge joints damage. In order to simulate the phenomenon of different descriptions of hinge joints damage degree by detection personnel, four different fuzzy detection intervals are given with damage degree 20% as the maximum credibility value, and eight different fuzzy detection intervals are given with damage degree 40% as the maximum reliability value. The division and numbering of the intervals are shown in Table 1.

**Table 1.** Division of damage degree interval.

| The Maximum Credibility is 20% | | The Maximum Credibility is 40% | | | |
|---|---|---|---|---|---|
| Interval Number | Damage Range | Interval Number | Damage Range | Interval Number | Damage Range |
| 1 | [15–25%] | 1 | [35–45%] | 5 | [15–65%] |
| 2 | [10–30%] | 2 | [30–50%] | 6 | [10–70%] |
| 3 | [5–35%] | 3 | [25–55%] | 7 | [5–75%] |
| 4 | [0–40%] | 4 | [20–60%] | 8 | [0–80%] |

Triangular, normal-type, and rectangular membership functions are selected for calculation in order to analyze the difference in reliability assessment results generated by fuzzy detection information of hinge joints described by different membership functions. In the normal membership function, $k = 1$ and $k = 2$ are two situations and are considered to represent two different confidence levels. Taking 1# slab as an example, the member failure probability of considering the fuzzy detection information of hinge joints under four kinds of hinge joint damage cases at different hinge joint damage intervals in Table 2 is calculated, and the failure probability of hinge joints in deterministic damage degree with

20% and 40% is calculated by the FOSM method based on the modified hinged–jointed plate method proposed in our research paper [47] and the response surface method. The member failure probability under the condition of considering fuzzy information and deterministic damage degree is compared, whose calculation results are shown in Figures 3 and 4.

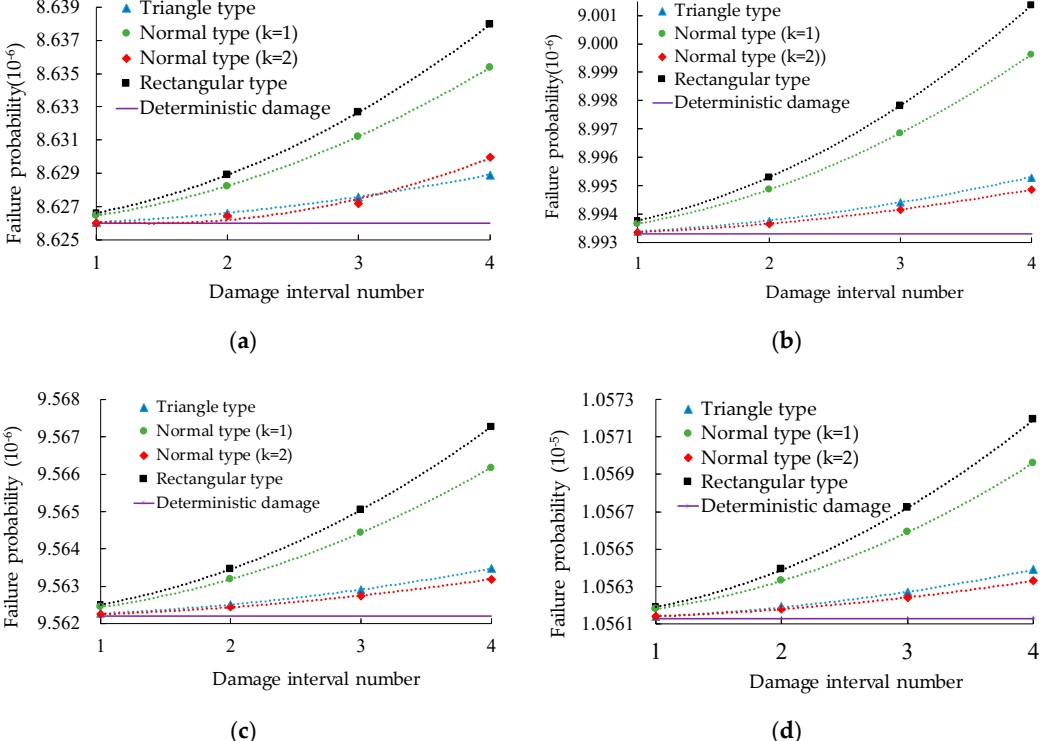

**Figure 3.** Failure probability of 1# slab in different hinge joint damage intervals under different membership functions. (The maximum credibility is 0.2): (**a**) Hinge joint damage case 1; (**b**) hinge joint damage case 2; (**c**) hinge joint damage case 3; (**d**) hinge joint damage case 4.

As can be seen from Figures 3 and 4, when the fuzzy detection information of hinge joint damage is expressed with different membership functions, the failure probability of #1 slab in all cases presents an increasing trend with the increase in the range of the damage interval, and the smaller the damage range is, the closer the failure probability is to failure probability corresponding to the degree of deterministic damage of hinge joints, which is in accordance with the objective law, so the accuracy of the proposed reliability analysis method considering the fuzzy detection of hinge joints damage is verified.

In addition, the change rate of failure probability varies greatly with the increase in interval range when different membership functions are used to deal with the damage interval of hinge joints.

The deviation between member failure probability and deterministic damage failure probability obtained by using different membership functions is the rectangle membership function, normal membership function ($k = 1$), triangle membership function, and normal membership function ($k = 2$), from large to small, which is mainly due to the different credibilities of different membership functions to the damage degree in the damage interval. As the rectangular membership function has the same damage degree credibility in the damage interval, the failure probability deviation is the largest. Therefore, the higher the detection level of the engineer, the smaller the detection range given, and the greater the credibility degree of the failure probability result in the detection of hinge joint damage.

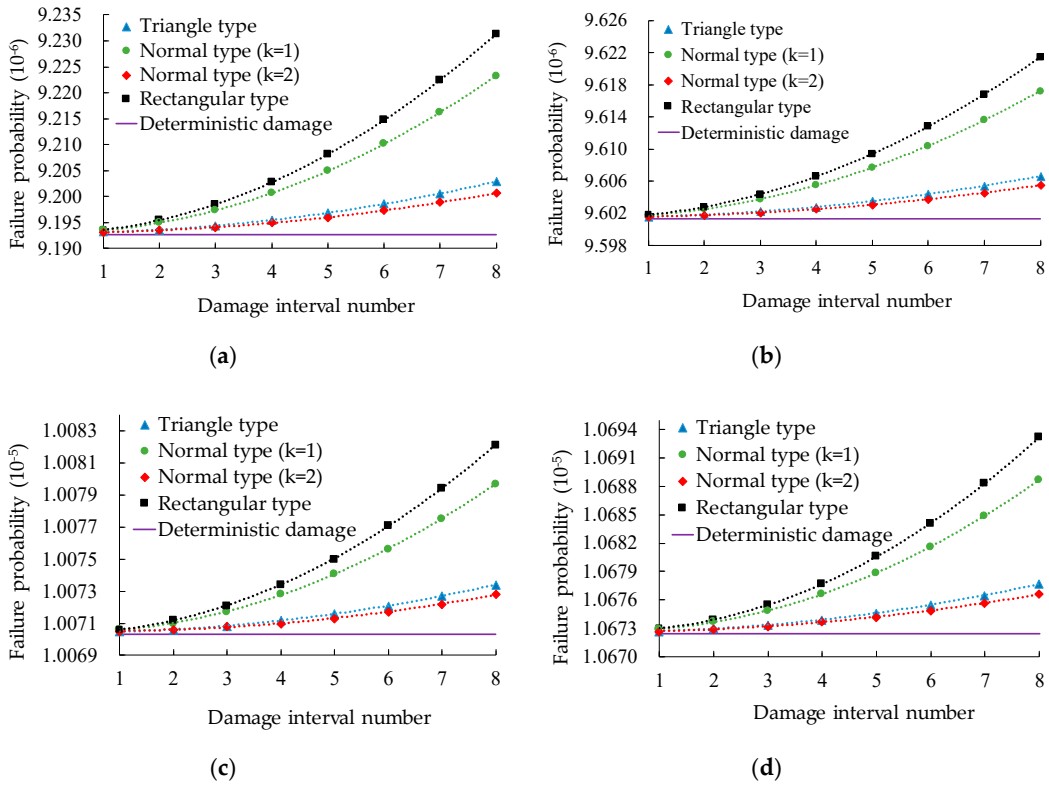

**Figure 4.** Failure probability of 1# slab in different hinge joint damage intervals under different membership functions. (The maximum credibility is 0.4): (**a**) Hinge joint damage case 1; (**b**) hinge joint damage case 2; (**c**) hinge joint damage case 3; (**d**) hinge joint damage case 4.

### 4.2. System Reliability Analysis

Because there are many failure modes in actual engineering, and there is a certain correlation among them, how to consider the correlation between failure modes is very important to the reliability theory of the structural system. Copula theory is introduced to analyze the correlation of failure modes of the structural system in this paper, and each Copula function has its own characteristics during calculation, so it is necessary to select the appropriate Copula function. First, the failure credibility of each member should be determined, as shown in Table 2.

**Table 2.** Failure credibility *Beli* of each member.

| $Bel_1$ | $Bel_2$ | $Bel_3$ | $Bel_4$ | $Bel_5$ |
|---------|---------|---------|---------|---------|
| 1.000   | 0.7793  | 0.7635  | 0.7793  | 1.000   |

For the optimization of the Copula function in this paper, the Akaike information criterion (AIC) and Bayesian information criterion (BIC) commonly used in engineering are adopted, that is, the Copula function with minimum AIC and BIC values is used as the optimal Copula function to fit the original observation data [48]. The commonly used Gaussian Copula, Clayton Copula, Gumbel Copula, and Frank Copula are selected as the candidate copula functions in this paper, and the corresponding Copula parameters, as well as the AIC and BIC values, are calculated, as shown in Tables 3 and 4.

**Table 3.** Parameter values of candidate copula functions.

| | Copula Parameters θ | | | |
|---|---|---|---|---|
| | **Gaussian** | **Clayton** | **Gumbel** | **Frank** |
| $Z_1Z_2$ | 0.9638 | 5.598 | 5.268 | 20.84 |
| $Z_1Z_3$ | 0.9635 | 5.567 | 5.319 | 20.72 |
| $Z_1Z_4$ | 0.9625 | 5.523 | 5.100 | 20.25 |
| $Z_1Z_5$ | 0.9567 | 5.22 | 4.84 | 19.13 |
| $Z_2Z_3$ | 0.9675 | 5.835 | 5.524 | 22.15 |
| $Z_2Z_4$ | 0.9673 | 6.02 | 5.484 | 21.58 |
| $Z_2Z_5$ | 0.9632 | 5.534 | 5.166 | 20.71 |
| $Z_3Z_4$ | 0.9664 | 5.76 | 5.516 | 21.64 |
| $Z_3Z_5$ | 0.9642 | 5.646 | 5.314 | 20.99 |
| $Z_4Z_5$ | 0.9647 | 6.011 | 5.23 | 21.00 |
| $Z_1Z_2Z_3$ | 0.965 | 5.236 | 5.158 | 20.26 |
| $Z_2Z_3Z_4$ | 0.9671 | 5.454 | 5.269 | 20.81 |
| $Z_3Z_4Z_5$ | 0.9651 | 5.443 | 5.139 | 20.18 |

**Table 4.** Akaike information criterion (AIC) and Bayesian information criterion (BIC) values of candidate copula functions.

| | **Gaussian** | | **Clayton** | | **Gumbel** | | **Frank** | |
|---|---|---|---|---|---|---|---|---|
| | **AIC** | **BIC** | **AIC** | **BIC** | **AIC** | **BIC** | **AIC** | **BIC** |
| $Z_1Z_2$ | −2630.94 | −2626.03 | −2060.63 | −2055.73 | −2504.95 | −2500.04 | −2411.98 | −2407.07 |
| $Z_1Z_3$ | −2623.55 | −2618.64 | −2045.51 | −2040.60 | −2525.52 | −2520.62 | −2400.81 | −2395.90 |
| $Z_1Z_4$ | −2595.22 | −2590.31 | −2047.60 | −2042.70 | −2442.23 | −2437.33 | −2373.22 | −2368.32 |
| $Z_1Z_5$ | −2455.64 | −2450.73 | −1955.75 | −1950.84 | −2334.87 | −2329.96 | −2255.88 | −2250.97 |
| $Z_2Z_3$ | −2738.17 | −2733.26 | −2119.30 | −2114.39 | −2605.01 | −2600.11 | −2516.50 | −2511.59 |
| $Z_2Z_4$ | −2730.31 | −2725.40 | −2176.92 | −2172.01 | −2580.07 | −2575.16 | −2473.04 | −2468.13 |
| $Z_2Z_5$ | −2615.39 | −2610.48 | −2055.53 | −2050.62 | −2465.32 | −2460.41 | −2401.07 | −2396.16 |
| $Z_3Z_4$ | −2703.77 | −2698.86 | −2106.95 | −2102.05 | −2592.93 | −2588.02 | −2475.32 | −2470.41 |
| $Z_3Z_5$ | −2642.35 | −2637.45 | −2075.89 | −2070.98 | −2517.99 | −2513.08 | −2421.48 | −2416.57 |
| $Z_4Z_5$ | −2655.93 | −2651.02 | −2163.67 | −2158.76 | −2488.56 | −2483.65 | −2426.43 | −2421.52 |
| $Z_1Z_2Z_3$ | −5603.47 | −5598.56 | −4338.82 | −4333.91 | −5315.29 | −5310.39 | −5121.51 | −5116.60 |
| $Z_2Z_3Z_4$ | −5726.49 | −5721.59 | −4467.73 | −4462.83 | −5400.57 | −5395.66 | −5215.75 | −5210.84 |
| $Z_3Z_4Z_5$ | −5611.87 | −5606.96 | −4455.50 | −4450.59 | −5280.78 | −5275.87 | −5106.59 | −5101.68 |

As can be seen from Table 4, the Gaussian Copula is determined as the most appropriate Copula function among the three failure criteria according to AIC and BIC minimum principles. Then, the combined failure probability of members considering the failure credibility under three failure criteria is calculated, combined with Equation (15) based on the Gaussian Copula, as demonstrated in Table 5.

**Table 5.** Joint failure probability considering failure credibility.

| | Joint Failure Probability $P_{f,Bel}$ | | |
|---|---|---|---|
| $Z_1Z_2$ | $1.069 \times 10^{-6}$ | $Z_3Z_4$ | $5.604 \times 10^{-7}$ |
| $Z_1Z_3$ | $9.316 \times 10^{-7}$ | $Z_3Z_5$ | $9.347 \times 10^{-7}$ |
| $Z_1Z_4$ | $1.062 \times 10^{-6}$ | $Z_4Z_5$ | $1.074 \times 10^{-6}$ |
| $Z_1Z_5$ | $4.016 \times 10^{-6}$ | $Z_1Z_2Z_3$ | $5.438 \times 10^{-7}$ |
| $Z_2Z_3$ | $5.679 \times 10^{-7}$ | $Z_2Z_3Z_4$ | $4.240 \times 10^{-7}$ |
| $Z_2Z_4$ | $6.145 \times 10^{-7}$ | $Z_3Z_4Z_5$ | $5.445 \times 10^{-7}$ |
| $Z_2Z_5$ | $1.066 \times 10^{-6}$ | | |

The system reliability of three different failure criteria considering member failure credibility is calculated. The above analysis process is adopted to calculate the system reliability of three different failure criteria without considering the member failure credibility, whose results are shown in Figure 5.

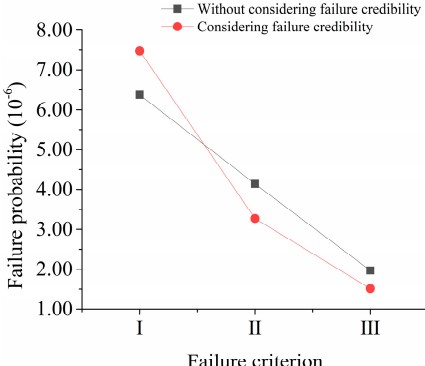

**Figure 5.** System reliability under different failure criteria considering failure credibility and not considering failure credibility.

It can be seen from Figure 5 that the failure credibility of members has an impact on the system reliability of the three failure criteria. Compared to not considering member failure credibility, the failure probability of failure criterion I increases while the failure probability of failure criterion II and III decreases when considering member credibility.

Because the above analysis does not consider hinge joint damage, the system reliability that considers both the member failure credibility and the fuzzy detection information of hinge joint damage degree is analyzed according to the reliability evaluation method considering the fuzzy detection information of hinge joint damage degree proposed in this paper. The corresponding four damage degree intervals in Table 2 when the damage degree of hinge joints is 0.2 as the maximum reliability value are selected. Taking case 1 as an example, the system reliability of different failure criteria is analyzed based on Copula theory, whose results are shown in Figure 6.

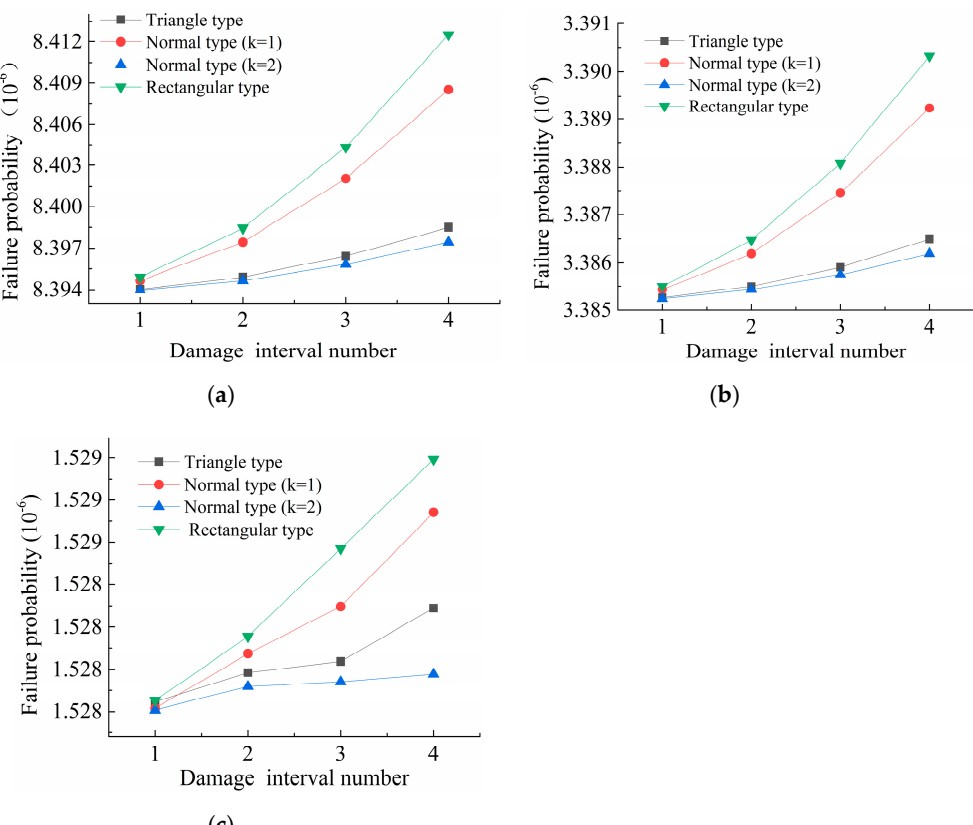

**Figure 6.** System reliability considering fuzzy detection information of the hinge joint damage degree and member failure credibility: (**a**) failure criterion I; (**b**) failure criterion II; (**c**) failure criterion III.

In the reliability evaluation of the system under different failure criteria, the failure probability of the system corresponding to different membership functions is basically consistent when the limit range of the hinge joint damage degree interval is minimum, and the failure probability gradually increases with the damage degree interval range. When the damage degree of hinge joints is expressed with three different membership functions, the change rate of the system failure probability corresponding to different membership functions is different as the damage range increases, and the change law is basically consistent with the change law of the member reliability. In practical engineering, more accurate evaluation results can be obtained by reducing the error caused by detection.

## 5. Conclusions

A reliability evaluation method of hinged slab beam bridges is proposed in this paper, which takes into account the fuzzy detection information of the damage degree of hinge joints and the failure reliability of the member. Firstly, the membership function is used to carry out equivalent randomization for fuzzy detection information of the damage degree of the hinge joints. Secondly, the reliability of a single member considering the fuzzy detection information of hinge joint damage is calculated. Then, the reliability of the structural system considering member failure credibility under three failure criteria is calculated. Finally, the applicability of the method is verified by taking a reinforced-concrete simply supported hollow slab beam bridge as a numerical example, and the following conclusions are drawn:

1.  Triangle, normal, and rectangle membership functions in fuzzy theory are used to express the damage degree of hinge joints in this paper, and the damage degree of hinge joints is transformed into a random variable. Numerical examples show that the equivalent randomization method of the membership function proposed is feasible.
2.  When different membership functions are used to express the damage interval, the change rate of failure probability varies greatly with the increase in the damage interval. With the reduction in the damage interval, the failure probability corresponding to the three membership functions is closer to the failure probability corresponding to the deterministic hinge joint damage degree, which indicates that the reliability evaluation method considering the fuzzy detection information of hinge joint damage proposed in this paper conforms to the objective law, and the rationality and accuracy of the method are verified.
3.  There is a deviation in the joint failure probability between considering member failure credibility and without considering member failure credibility under different criteria calculated based on Copula theory, and member failure credibility has an impact on the system reliability. Therefore, it is meaningful to consider the failure credibility in the system reliability evaluation method.
4.  Furthermore, on the basis of this study, the authors will carry out the fragility analysis and reliability evaluation method of medium- and small-span bridges under natural disasters (earthquakes, floods).

**Author Contributions:** Conceptualization, G.T. and L.W.; methodology, G.T. and Q.K.; software, X.W.; validation, L.W., H.L. and X.W.; formal analysis, G.T. and L.W.; investigation, Q.K. and L.W.; writing—original draft preparation, G.T. and Q.K.; writing—review and editing, L.W.; funding acquisition, G.T. and H.L. All authors have read and agreed to the published version of the manuscript.

**Funding:** This research was funded by the National Natural Science Foundation of China (grant number 51978309), Science and Technology Project of Education Department of Jilin Province (grant numberJJKH20190150KJ, JJKH20180152KJ), Scientific and Technological Project of Science and Technology Department of Jilin Province (grant number 20190303052SF), Jilin Province Development and Reform Commission Project (grant number 2019C041-5) and Jilin Transportation Science and Technology Popularized Project (grant number 2020-3-2).

**Acknowledgments:** The authors would like to thank the anonymous reviewers for their constructive suggestions and comments toward improving the quality of the paper.

**Conflicts of Interest:** The authors declare no conflict of interest.

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
