# Peer review of "Reliability Evaluation of Hinged Slab Bridge Considering Hinge Joints Damage and Member Failure Credibility"

_applsci, doi:10.3390/app10144824_

Round 1

Reviewer 1 Report

The submitted article can be published after the following major changes are made. 

  1. In introduction many studies are reported to be relevant to this study, however, I could not find all of them well suited to this study and without pointing on any I suggest the authors to remove the irrelevant studies from the list. Also, the old studies are needed to be so many, you can keep only those which are quite important and must be explained. There are many other relevant studies which are missing in the introduction and can be added to provide better background to readers. 

Pagnoncelli, A. P., & Miguel, L. F. F. (2019). Methodology to Obtain Dynamic Response of Road Bridges Considering Bridge–Vehicle Interactions. Practice Periodical on Structural Design and Construction24(3), 04019010.

Matin, A., Elias, S., & Matsagar, V. (2020). Distributed multiple tuned mass dampers for seismic response control in bridges. Proceedings of the Institution of Civil Engineers-Structures and Buildings173(3), 217-234.

Hou, G., & Chen, S. (2017). Bent Connection Options for Curved and Skewed SMC Bridges in Low-to-Moderate Seismic Regions. Practice Periodical on Structural Design and Construction22(4), 04017011.

Li, J., & Chen, G. (2011). Method to compute live-load distribution in bridge girders. Practice Periodical on Structural Design and Construction16(4), 191-198.

Pipinato, A., & Modena, C. (2010). Structural analysis and fatigue reliability assessment of the Paderno bridge. Practice Periodical on Structural Design and Construction15(2), 109-124.

Carroll, C., & Juneau, A. (2015). Repair of Concrete Bridge Deck Expansion Joints Using Elastomeric Concrete. Practice Periodical on Structural Design and Construction20(3), 04014038.

Wang, H., Xie, C., Liu, D., & Qin, S. (2019). Continuous Reinforced Concrete Rigid-Frame Bridges in China. Practice Periodical on Structural Design and Construction24(2), 05019002.

Elias, S., & Matsagar, V. (2017). Effectiveness of tuned mass dampers in seismic response control of isolated bridges including soil-structure interaction. Latin American Journal of Solids and Structures14(13), 2324-2341.

Xie, Y., Yang, H., Zuo, Z., Sirotiak, T. L., & Yang, M. (2018). Optimal Steel Section Length of the Composite Rigid-Frame Bridge. Practice Periodical on Structural Design and Construction23(3), 05018001.

Mahamid, M., Ozevin, D., Torra-Bilal, I., Kabir, M., Mastny, S., Khudeira, S., ... & Zroka, D. (2019). Structural Design and Inspectability of Highway Bridges. Practice Periodical on Structural Design and Construction24(3), 06019002.

2. Using a flow chart to show the novelty and overall process can help the readers to understand the study in a better way. 

3. Mostly researchers are showing Failure probability  by fragility curves but here it is not the standard form, I would suggest in standard way, or keep both for better understanding of the readers. 

4. I could not find the details of the bridge and location of hinges, nor the figure is provided. This is not a good presentation to jump to discussion without knowing the structure. 

Author Response

Response to Reviewer 1 Comments

Point 1: In introduction many studies are reported to be relevant to this study, however, I could not find all of them well suited to this study and without pointing on any I suggest the authors to remove the irrelevant studies from the list. Also, the old studies are needed to be so many, you can keep only those which are quite important and must be explained. There are many other relevant studies which are missing in the introduction and can be added to provide better background to readers.

 Response 1: Thank you very much for your meaningful comments, which will make our paper more rigorous. All of studies in your suggested list are mainly bridge structure analysis, which are of great significance to our paper. So after reading these papers carefully, I decided to cite them in Introduction, and we also remove some irrelevant or old studies and replaced them with more appropriate studies. The detailed modification is listed as follows:

Line 33: With the extension of the service period, the service function of the bridge will gradually deteriorate under the repeated action of external environment and external load (such as floods, earthquakes, and vehicles, etc.) [1-7]. Therefore, in addition to optimizing the design of the bridge, it is of vital importance to accurate judgment of the state of the bridge, which can not only provide a reasonable basis for the bridge maintenance decision, but also ensure its safe operation [8-14]. Hinged slab bridges are widely used in medium and small span bridges due to their simple structure and convenient construction, whose one of the biggest defects is hinge joint damage, so it is of great significance to evaluate the status of hinged slab bridge under hinge joint damage [15-18].

Line 86: In terms of calculating the failure probability of the system, interval estimation method (mainly including narrow bounds method and wide bounds method) and probabilistic network technique method are often used to calculate the failure probability of the system [34].

The list of added references:

[3] Pagnoncelli, A. P.; Miguel, L. F. Methodology to Obtain Dynamic Response of Road Bridges Considering Bridge–Vehicle Interactions. Pract. Period. Struct. Des. Constr. 2019, 24(3), 04019010.

[4] Matin, A., Elias, S., & Matsagar, V. (2020). Distributed multiple tuned mass dampers for seismic response control in bridges. Proc. Inst. Civil Eng.-Struct. Build. 2020, 173(3), 217-234.

[5] Hou, G.; Chen, S. Bent Connection Options for Curved and Skewed SMC Bridges in Low-to-Moderate Seismic Regions. Pract. Period. Struct. Des. Constr. 2017, 22(4), 04017011.

[6] Elias, S.; Matsagar, V. Effectiveness of tuned mass dampers in seismic response control of isolated bridges including soil-structure interaction. Lat. Am. J. Solids Struct. 2017, 14(13), 2324-2341.

[7] Li, J.; Chen, G. Method to compute live-load distribution in bridge girders. Pract. Period. Struct. Des. Constr. 2011, 16(4), 191-198.

[11] Pipinato, A.; Modena, C. Structural analysis and fatigue reliability assessment of the Paderno bridge. Pract. Period. Struct. Des. Constr. 2010, 15(2), 109-124.

[12] Mahamid, M.; Ozevin, D.; Torra-Bilal, I.; et al. Structural Design and Inspectability of Highway Bridges. Pract. Period. Struct. Des. Constr. 2019, 24(3), 06019002.

[13] Wang, H.; Xie, C.; Liu, D.; et al. Continuous Reinforced Concrete Rigid-Frame Bridges in China. Pract. Period. Struct. Des. Constr. 2019, 24(2), 05019002.

[14] Xie, Y.; Yang, H.; Zuo, Z.; et al. Optimal Steel Section Length of the Composite Rigid-Frame Bridge. Pract. Period. Struct. Des. Constr. 2018, 23(3), 05018001.

[18] Carroll, C.; Juneau, A. Repair of Concrete Bridge Deck Expansion Joints Using Elastomeric Concrete. Pract. Period. Struct. Des. Constr. 2015, 20(3), 04014038.

[34] Wang, H.; Zhang, X.; Li, Q.; et al. Recursive Method for Distribution System Reliability Evaluation. Energies. 2018, 11(10), 2681.

In addition, the number of references in the original paper has been adjusted due to the addition of these added references.

Point 2: Using a flow chart to show the novelty and overall process can help the readers to understand the study in a better way.

Response 2: Thank you very much for your meaningful comments, which will make our paper more rigorous. We added a flow chart of the reliability evaluation method proposed in this paper to show the novelty and overall process. The added flow chart is shown as follows:

Line 146: The specific process of reliability evaluation method proposed in this paper is shown in Figure 1.

Figure 1. Flow chart of the reliability evaluation method proposed in this paper

Point 3: Mostly researchers are showing Failure probability by fragility curves but here it is not the standard form, I would suggest in standard way, or keep both for better understanding of the readers

Response 3: Thank you very much for your careful reading and significant comments. The fragility curve is commonly used in structural performance evaluation that consider the uncertainties under extreme actions (such as earthquakes, floods, wind loads, etc.), which can reflect the structural failure probability under external loads of different levels and strengths. However, the purpose of this paper is to discuss the reliability of hinged slab bridge considering the fuzzy detection information of hinge joint damage and the member failure reliability under the certain vehicle external load, and the abscissa of Figures 3, 4 represents different fuzzy detection intervals, not the degree of the damage, so the fragility curve analysis is not used in this paper. Of course, the fragility curve is very suitable and easy to understand in the reliability analysis of variable external loads. With the further study of our ongoing project, we will use fragility analysis method to evaluate the reliability of medium and small span bridges under earthquake and flood. We have added the future research in Line 443:

"Furthermore, on the basis of this study, the authors will carry out the fragility analysis and reliability evaluation method of medium and small span bridges under natural disasters (earthquakes, floods)."

Point 4: I could not find the details of the bridge and location of hinges, nor the figure is provided. This is not a good presentation to jump to discussion without knowing the structure.

Response 4: Thank you for your critical comments! The bridge material properties were the same as last research paper [47], we cited the reference directly without making a detailed introduction due to the limitation of length, we are sorry for not giving the sufficient details about numerical simulation example of RC bridge. Therefore we have added more details about numerical simulation example in Line 312, the detailed modification is listed as follows:

The bridge is a single span reinforced concrete hollow slab beam, which is composed of five hollow slabs. The total length of the bridge is 10.0m meters and the total width of the bridge deck is 7.5m. The main beam is constructed with concrete of grade C50, reinforced with HRB335 and R235 steel bars. The material properties of bridge, as well as the statistical parameters and probability distribution types of random variables of main slab resistance, dead load effect and live load effect are the same as those of reference [47]. The profile diagram of the bridge and the location of hinges are shown in Figure 2.

Figure.2 Bridge profile diagram.

Reviewer 2 Report

Dear Editor,

Thanks for your invitation to review Manuscript entitled: Reliability evaluation of hinged slab bridge considering the fuzzy detection information of hinge joints damage and the failure credibility of

Component.

I suggest minor corrections for this article as follows:

  • Consider shorter title for the article, remove detailed phrases.
  • Improve the language of the paper for more clarity, use words with more technical meanings. Replace words such as fuzzing, disease and similar words.
  • Include sources of the equations used in the article.
  • Include more details about numerical simulation example of RC bridge. Such as material properties, external loads, sketches and more similar details.
  • Explain how to calculate the deterministic damage degree as in Figures 1 and 2.

Regards,

Author Response

Response to Reviewer 2 Comments

Point 1: Consider shorter title for the article, remove detailed phrases.

Response 1: Thanks very much to point out this important modification. The title “Reliability evaluation of hinged slab bridge considering the fuzzy detection information of hinge joints damage and the failure credibility of component” is changed into “Reliability Evaluation of Hinged Slab Bridge Considering Hinge Joints Damage and Member Failure Credibility”

Point 2: Improve the language of the paper for more clarity, use words with more technical meanings. Replace words such as fuzzing, disease and similar words.

Response 2: Thanks very much for your comments. The language, spelling mistakes and sentences throughout the whole manuscript have been checked carefully. Some changes are listed as follows:

  1. "the Failure Credibility of Component" is changed into "the Member Failure Credibility" and "component failure" is changed into "member failure " in the whole paper.
  2. Line 13 "main disease" is changed into " main defect".
  3. Lines 20, 221 "improved hinge-jointed plate method" is changed into "modified hinge-jointed plate method".
  4. Line 39 "biggest diseases" is changed into "biggest defects".

Point 3: Include sources of the equations used in the article.

Response 3:

We are very sorry that some of the sources of the equations are not clearly stated. Equation (16) is derived on the basis of previous research paper [48] after considering the failure credibility, therefore we have added Equation (15) in this paper which is given in [48] to better illustrate the source of the equation in this paper. Equations (17-19) are derivation of the Equations (16) based on research paper [48], and we have made a citation in the manuscript. Other equations in this paper are all put forward by the authors. The detailed modification are listed as follows:

  1. Line 276:

“The joint failure probability of multiple members simultaneous failure as follows according to the Copula theory [48]:

              (15)

where  denotes the limit state equation of i-th members,  represents the failure probability of the i-th member, respectively; C(·) expresses copula function.

 On this basis, member failure credibility is considered in this paper, the joint failure probability of n  members established by Copula function is expanded in Equation ( 16).”

         ( 16)

  1. Line 287: “The authors of this paper proposed a structural system reliability assessment method considering three kind of system failure criteria based on Copula function calculation [48], Equation (15) is adopted in this paper, to improve the system reliability assessment method previously proposed based on the failure credibility, and the system reliability evaluation method considering component member failure credibility under failure criterion І, П, Ш is formed respectively, which are derived as Equations (16) ~ (18).”

Point 4: Include more details about numerical simulation example of RC bridge. Such as material properties, external loads, sketches and more similar details.

Response 4: Thank you for your critical comments! The bridge material properties were the same as last research paper [47], we quoted the reference directly without making a detailed introduction due to the limitation of length, we are sorry for not giving the sufficient details about numerical simulation example of RC bridge. Therefore we have added more details about numerical simulation example in Line 312; the detailed modification is listed as follows:

The bridge is a single span reinforced concrete hollow slab beam, which is composed of five hollow slabs. The total length of the bridge is 10.0m meters and the total width of the bridge deck is 7.5m. The main beam is constructed with concrete of grade C50, reinforced with HRB335 and R235 steel bars. The material properties of bridge, as well as the statistical parameters and probability distribution types of random variables of main slab resistance, dead load effect and live load effect are the same as those of reference [47]. The profile diagram of the bridge and the location of hinges are shown in Figure 2.

Figure.2 Bridge profile diagram.

Point 5: Explain how to calculate the deterministic damage degree as in Figures 1 and 2.

Response 5: Thanks very much for your instructive comments and we are very grateful to reviewer #2 for raising this question. The deterministic damage degree in Figures 1 and 2 indicates that the maximum credibility of the hinge joint damage degree is 20% and 40%, respectively. The calculation process is as follows: the transverse distribution coefficient is calculated by the modified hinged-jointed plate method proposed in our research paper [47], then the transverse distribution coefficient is substituted into the load effect model, and the first order second moment method is used to calculate the failure probability of hinge joint with deterministic damage degree. We are sorry that the explanation of the deterministic damage degree in our manuscript is not detailed, so the detailed modification is listed as follows:

Line 337: “Taking 1# slab as an example, the component failure probability of considering the fuzzy detection information of hinge joints under four kinds of hinge joints damage cases at different hinge joints damage intervals in Table 2 is calculated respectively, and the failure probability is compared with the failure probability corresponding to the deterministic damage degree, whose results are shown in Figure 1 and Figure 2.” is changed into “Taking 1# slab as an example, the member failure probability of considering the fuzzy detection information of hinge joints under four kinds of hinge joints damage cases at different hinge joints damage intervals in Table 2 is calculated respectively. And the failure probability of hinge joint in deterministic damage degree with 20% and 40% is calculated respectively by FOSM method based on the modified hinged-jointed plate method proposed in our research paper [47] and response surface method. The member failure probability under the condition of considering fuzzy information and deterministic damage degree is compared, whose calculation results are shown in Figure 1 and Figure 2.”

Reviewer 3 Report

The article is sufficiently novel and interesting to warrant publication and it adheres to the journal's standards. The article is clearly laid out. All the key elements are present: abstract, introduction, methodology, results, discussion and conclusions. The title clearly describes the article and the abstract reflects the content of the article. The introduction contains a brief description of the actual state-of-the-art, and clearly state the problem being investigated. The authors accurately explain what they discovered in the research. The claims in conclusion are supported by the results. The references are accurate. The reviewer recommends accepting the paper for publication provided that the Authors take into account the following comment:

- Some symbols in equations are not defined in the main text. Symbols must be defined before or after equations, when they are used first.

Author Response

Response to Reviewer 3 Comments

Point 1: The article is sufficiently novel and interesting to warrant publication and it adheres to the journal's standards. The article is clearly laid out. All the key elements are present: abstract, introduction, methodology, results, discussion and conclusions. The title clearly describes the article and the abstract reflects the content of the article. The introduction contains a brief description of the actual state-of-the-art, and clearly state the problem being investigated. The authors accurately explain what they discovered in the research. The claims in conclusion are supported by the results. The references are accurate. The reviewer recommends accepting the paper for publication provided that the Authors take into account the following comment:

- Some symbols in equations are not defined in the main text. Symbols must be defined before or after equations, when they are used first. 

Response 1: We are grateful for your recognition of our paper, and we have carefully proofread the logic and spelling of the whole manuscript. Thank you very much for your careful reading and significant comments. We are sorry for our negligence in not defining some symbols in equations, so the detailed modifications are listed as follows:

  1. “where is fuzzy variable.” is added in Line 173.
  2. “ represents the random variable obtained from the transformation of the fuzzy detection information of the i-th hinge joint damage; the remaining parameters are the undetermined coefficients.” is added in Line 231.

Round 2

Reviewer 1 Report

I accept the revised version and recommend it for publication.

Reviewer 2 Report

Dear Editor,

I read the Manuscript and I find that the authors had addressed most of the reviewers's comments.

I think it can be published at this time.

Regards,